# The Role of Immunoglobulin G (IgG), IgA and IgE—Antibodies against *Helicobacter pylori* in the Development of Oxidative Stress in Patients with Chronic Gastritis

**DOI:** 10.3390/biomedicines10082053

**Published:** 2022-08-22

**Authors:** Olga Valentinovna Smirnova, Alexander Alexandrovich Sinyakov, Eduard Vilyamovich Kasparov

**Affiliations:** Laboratory of Clinical Pathophysiology, Federal Research Center “Krasnoyarsk Science Center” of the Siberian Branch of the Russian Academy of Sciences, Research Institute of Medical Problems of the North, 660022 Krasnoyarsk, Russia

**Keywords:** *H. pylori*, oxidative stress, immune response

## Abstract

**Aim**: To study the predominant serum responses (antibodies IgG, IgA, IgE) against *H. pylori* in relation to the indicators of the system “lipid peroxidation–antioxidant system” in various pathogenetic variants of chronic gastritis (CG). **Materials and Methods**: Sixty patients with CG, 33 patients with chronic atrophic gastritis (CAG) and 31 patients with chronic allergic gastritis (CALG) were examined. The values of the system of lipid peroxidation and antioxidant protection in plasma were determined in the serum of patients using a spectrophotometric method. Statistical data processing was carried out using the Statistica 7.0 software package (StatSoft, Tulsa, OK, USA). **Results**: With serum responses “antibodies IgG > IgA” and “high concentrations of IgE antibodies”, we found unidirectional changes in the form of an increase in the amount of diene conjugates, malondialdehyde and an increase in the activity of all enzymes: superoxide dismutase, catalase, glutathione-S-transferase and glutathione peroxidase. With a serum response with low concentrations of IgG, IgA antibodies, multidirectional changes were found in the form of an increase in the amount of diene conjugates, malondialdehyde and a decrease in the activity of all enzymes: superoxide dismutase, catalase, glutathione-S-transferase and glutathione peroxidase relative to the control group. **Conclusions**: The obtained data testify to the balance of lipid peroxidation and antioxidant system processes and depend on the characteristics of the immune response to *H. pylori* infection.

## 1. Introduction

*Helicobacter pylori* (*H. pylori*) is a spiral Gram-negative rod that causes infiltration of the gastric mucosa by neutrophils, macrophages, T- and B-lymphocytes. The developing immune and inflammatory response does not eliminate the bacterial infection, allowing the pathogen to persist and making the host susceptible to the complications resulting from chronic inflammation. The prognosis will depend on changes in the gastric mucosa as a result of exposure to pathogenic factors, chronicization of the immunoinflammatory process in the stomach leads to the development of gastritis with glandular atrophy, intestinal metaplasia, dysplasia, etc., in various forms. In addition, there is a group of patients with chronic gastritis (CG) in whom *H. pylori* causes a hyperergic immune response with the development of allergies. The role of immune responses in the pathogenesis of *H. pylori* and chronic gastritis (CG) is an area of rapidly developing research [1,2,3,4]. It is known that the determination of serum specific antibodies plays an important role in monitoring *Helicobacter pylori* infection [5,6]. However, the significance of *Helicobacter pylori* infection and indicators of serum-specific immunoglobulins in various types of chronic gastritis have not been fully studied. Oxidative stress initiated by *H. pylori* can cause DNA destruction and may itself be the direct cause of neoplastic transformation of epithelial cells of the gastric mucosa, causing the formation of gastric adenocarcinoma in the future [7].

In this regard, the purpose of our study was to study the predominant serum responses (antibodies IgG, IgA, IgE) against *H. pylori* in conjunction with the indicators of the system “lipid peroxidation–antioxidant system” in various pathogenetic variants of CG.

## 2. Materials and Methods

### 2.1. Subjects

Patients admitted for examination to the therapeutic department of the clinic of the FGBNU FRC KSC Research Institute of Medical Problems of the North of the Siberian Branch of the Russian Academy of Sciences in Krasnoyarsk, from 2019 to 2021. Of all those admitted, patients were selected with chronic gastritis (CG) in the amount of 60 people, chronic atrophic gastritis (CAG) in the amount of 33 people and chronic allergic gastritis (CALG) in the amount of 31 people. The study included only primary patients with a verified diagnosis without a history of the use of proton pump inhibitors and non-steroidal anti-inflammatory drugs for the purity of the experiment. All patients included in the study were infected with *Helicobacter pylori*. The study did not include patients with peptic ulcer, after surgery, or taking antibiotics for the last 3 months. A total of 124 patients were included in the study, and the control group consisted of 60 apparently healthy blood donors (Figure 1).

This study was approved by the local ethics committee at the clinic of the Research Institute of Medical Problems of the North (protocol № 10; accessed on 16 September 2019). The ethical principles of the Declaration of Helsinki of the World Medical Association were observed in the work with the examined patients. All study participants were informed of the purpose and design of the study, and each patient signed an informed consent form to participate in the study.

### 2.2. Endoscopic Examination of the Stomach, Histological Examination, Testing for Helicobacter pylori and Sampling of Gastric Juice

All patients included in the study underwent three types of testing for the presence of *Helicobacter pylori*: histological examination, rapid urease test and culture. All biopsy material was analyzed by an experienced gastroenterologist. In order to prevent infection, the endoscope was washed and disinfected with an immersion detergent solution. In addition, serum IgG to *H. pylori* was measured by enzyme immunoassay (BIOHIT HealthCare, Helsinki, Finland). Specific antibodies to *Helicobacter pylori* of more than 30 EIU were regarded as a positive result of infection with *Helicobacter pylori*. All patients included in the study were proven to be infected with *Helicobacter pylori*. Histological confirmation of *H. pylori* infection was verified with modified Giemsa stain, positive threshold urease test (CLOtest; Delta West, Bentley, WA, Australia), positive *H. pylori* culture test, as well as ELISA method for measuring IgG to *H. pylori* in serum.

### 2.3. Collection of Gastric Juice and Measurement of Its pH

Approximately 5 mL of gastric juice was carefully taken from all patients from the gastric fundus by one doctor (V.V.). The gastric juice was transported as quickly as possible to the laboratory. The gastric juice was centrifuged (3000 rpm, 5 min), the supernatant was collected, and then the pH of the gastric juice was determined using a glass electrode (inoLab pH Level 1; WTW, Weilheim, Germany).

### 2.4. Determination of Serum Indicators of Pepsinogens

Serum was collected from all study patients on an empty stomach in the morning prior to endoscopy. Blood samples were centrifuged and stored at −70 °C. Serum levels of pepsinogen I and pepsinogen II were determined using ELISA kits (BIOHIT HealthCare, Helsinki, Finland). The ratio of pepsinogen I to pepsinogen II was calculated.

### 2.5. Serum IgA, IgG, IgM, IgE Levels

Serum was collected from all study patients on an empty stomach in the morning prior to endoscopy. Samples were centrifuged and stored at −70 °C. The concentration of IgM, IgG, and IgA antibodies against *Helicobacter pylori* was determined by ELISA for humans (SunRed Biotechnology Company, Shanghai, China) on a thermos scientific MULTISKAN FC ELISA analyzer (Thermo Fisher Scientific, Waltham, MA, USA). The concentration of total serum IgE immunoglobulin was determined by enzyme immunoassay using ELISA kit for humans ELISA (SunRed Biotechnology Company, Shanghai, China).

### 2.6. Lipid Peroxidation and Antioxidants

Plasma oxidative stress scores (MDA, SOD, CAT, GST, GPO). Indicators of lipid peroxidation and parameters of antioxidant protection were determined in the serum of patients using a spectrophotometric method on the device—Thermo SCIENTIFIC GENESYS 10vis (Thermo Fisher Scientific, Waltham, MA, USA).

#### 2.6.1. Determination of the Content of Diene Conjugates

The principle of the method is based on the intense absorption of conjugated diene structures of lipid hydroperoxides in the region of 232 nm. To calculate DC, the molar extinction coefficient was used:(1)$$K=2.2\;\times 10^{5}\;M^{-1}{\mathrm{cm}}^{-1}$$

The content of diene conjugates was expressed in µmol/L.

#### 2.6.2. Determination of Malondialdehyde Content

MDA is formed in lipid systems as a result of lipid peroxidation. When interacting with 2-thiobarbituric acid (TBA), a chromogen is formed with an absorption maximum in the red region of the visible spectrum at a wavelength of 532 nm. The calculation of the MDA content is carried out taking into account the molar extinction coefficient of the formed chromogen, equal to 1.56 × 10^5^ M^−1^ cm^−1^, and is expressed in µmol/L:(2)$$C=\frac{D532\times V_{p.c.}\times 1000}{V*\varepsilon *P*d}$$

#### 2.6.3. Determination of the Amount of Reduced Glutathione

The definition is based on the interaction of GSH with DTNBA (5,5′-dithio-bis-2-nitrobenzoic acid) to form a yellow-colored 2-nitro-5-thiobenzoate anion. The increase in the concentration of the yellow anion during this reaction was recorded spectrophotometrically at a wavelength of 412 nm. Samples are photometered before and after the addition of DTNBA.

#### 2.6.4. Determination of Glutathione-S-Transferase Activity

The activity of glutathione-S-transferase was measured by the rate of formation of glutathione-S-conjugates between GSH and 1-chloro-2,4-dinitrobenzene (CDNB). The increase in the concentration of conjugates during the reaction was recorded spectrophotometrically at a wavelength of 340 nm (maximum absorption of glutathione-S-CDNB).

Enzyme activity was calculated using the millimolar extinction coefficient for GS-CDNB at a wavelength of 340 nm, equal to 9.6 mM^−1^ cm^−1^, and expressed in micromoles of glutathione-S-conjugates formed per minute per 1 gram of Hb:(3)$$A=\frac{\Delta E/\min*V_{p.c.}*1000}{\varepsilon *V*Hb*d}$$

#### 2.6.5. Determination of Glutathione Peroxidase Activity

Glutathione peroxidase (GPO) catalyzes the reaction between glutathione (GSH) and tert-butyl hydroperoxide (HTB):

GSH + HTB = (GPO) = GSSO + reducedHTB(4)

Enzyme activity is assessed by the change in the GSH content in samples before and after incubation with a model substrate in the course of a color reaction with dithionitro (bis)benzoic acid (DTNBA). Measure the extinction of the experimental and control samples for CPK at a wavelength of 412 nm; zeroed in distilled water.

Activity is calculated by the formula
(5)$$A=\frac{\Delta D*V_{p.c.}*1000}{V*\varepsilon *Hb*d}$$

#### 2.6.6. Determination of Superoxide Dismutase Activity

The principle of the method is based on the inhibition of the adrenaline autoxidation reaction in an alkaline medium in the presence of superoxide dismutase (SOD).

The intensity of adrenaline autoxidation was judged by the dynamic increase in absorption at a wavelength of 347 nm, due to the accumulation of an oxidation product not previously described in the literature and the adrenochrome formation with an absorption maximum at 480 nm ahead of time.
(6)$$\mathrm{Unit}\;\mathrm{of}\;\mathrm{activity}\frac{\mathrm{SOD}}{\Gamma Hb}=\left(\frac{Ex-Eo}{Ex}\;\right)*\;\frac{100\%*F*V*1000}{50*v*d*C}$$
where
(7)$$\frac{Ex-Eo}{Ex}\;*\;\frac{100\%}{50}\;\mathrm{unit}\;\mathrm{of}\;\mathrm{activity}\;\mathrm{per}\;\mathrm{milliliter}\;\mathrm{of}\;\mathrm{plasma}$$

#### 2.6.7. Determination of Catalase Activity

The determination of catalase activity is based on the formation of a yellow-colored complex of hydrogen peroxide with ammonium molybdate that was not destroyed during the catalase reaction. Catalase activity is calculated by the formula:(8)$$A=\frac{\frac{\Delta A}{c}*V*f}{t*v*d*K*Hb*60}$$

#### 2.6.8. Definition Oxidative Stress Ratio (OSR)

The method of individual assessment of oxidative stress by calculating the integral coefficient according to the ratio of pro- and antioxidant factors (Figure 2), where i represents the levels of indicators of the examined patients, n represents levels of indicators of the comparative group, with OSR > 1, and the development of oxidative stress is recorded.

### 2.7. Statistical Analysis

Statistical data processing was carried out using the Statistica 7.0 software package (StatSoft, Tulsa, OK, USA). The analysis of the conformity of the type of distribution of the sign to the law of normal distribution was carried out using the Shapiro–Wilk test. When describing the sample, medians (Me) and interquartile range of percentiles (Q_1_–Q_3_) were calculated. The significance of differences between the indicators of independent samples was assessed by the Mann–Whitney test (*p* < 0.05). To determine the significance of differences in one trait depending on another, the chi-square test (χ^2^) was calculated.

## 3. Results

### 3.1. Baseline Patient Data

The mean age of the patients was 53 years (Table 1). In patients with chronic atrophic gastritis, a study of the gastric mucosa was carried out using endoscopic biopsy; atrophy of the gastric mucosa in the antrum was found in 28 patients and in the body of the stomach in 26 patients. According to the updated Sydney scoring system, 18 patients (52.9%) and 13 patients (36.1%) had a mild degree of CAG; the remaining patients had a higher degree of CAG in the antrum and on the body, respectively. The average pH value of gastric juice in patients with chronic gastritis is 3.23, CAG—4.54, CALG—2.46. Serum levels of PG I and PG II were confirmed in 124 patients, and the average level of PG I and PG II in patients with CG was 56.48 and 19.32 ng/mL. In patients with CAG, it was 93.17 and 34.80, and in patients with CALG, it was 66.33 and 27.56, respectively.

### 3.2. Distribution of Chronic Gastritis according to the Specifics of the Humoral Response to H. pylori Infection

During the first stage of the study, we studied the prevalence of *H. pylori* infection in various types of chronic gastritis and in the control group. The results of endoscopic examination of persons in the control group revealed infection with *H. pylori* in 71% of those examined with asymptomatic course.

The total number of examined patients was 124 (Table 2). In patients with chronic gastritis, the presence of *H. pylori* infection was detected in 78% of cases. In patients with CAG, the presence of this infection was detected in 91% of cases, and in patients with CALG, the presence of H. pylori was 83%. We obtained a different prevalence of *H. pylori* infection in different types of gastritis in primary patients and in the control group. Subsequently, only individuals with *H. pylori* infection, proven by endoscopic and serological methods, were taken into the study.

Later, we studied the characteristics of serum responses (IgG, IgA, IgE antibodies) against *H. pylori* in various types of chronic gastritis (Table 3).

In patients with chronic gastritis, 75% of patients with a serum response against *H. pylori* IgG > IgA were identified; 15% of patients with a serum response in the form of low concentrations of IgG, IgA; 10% of patients with a serum response in the form of an increase in IgE. In patients with CAG, 60% of patients with a serum response against *H. pylori* IgG > IgA were detected; 27% of patients with a serum response in the form of low concentrations of IgG, IgA; 13% of patients with a serum response in the form of an increase in IgE. In 91% of patients with CALG, a serum response in the form of an increase in IgE was detected, and against the background of this response, 5% of patients with a serum response against *H. pylori* IgG > IgA and 4% of patients with a serum response in the form of low concentrations of IgG, IgA were identified.

### 3.3. Indicators of Lipid Peroxidation–AOD in Chronic Gastritis, Depending on the Specific Humoral Response to H. pylori Infection

#### 3.3.1. The State of the System “Lipid Peroxidation–Antioxidant Defense System” in Various Types of Chronic Gastritis in the Presence of a Serum Response against *H. pylori* in the Form of “IgG > IgA”

At the next stage, we studied the indicators of the LPO–AOD system with the same serum response against *H. pylori* in various types of chronic gastritis. When studying the lipid peroxidation system, the concentrations of primary (diene conjugates) and final (malonic dialdehyde) products of lipid peroxidation formed at various stages of the free radical chain reaction were studied. The activity of antioxidant defense was judged by the content of its main components (superoxide dismutase, catalase, glutathione-S-transferase, glutathione peroxidase, ceruloplasmin).

We assessed the state of the system “lipid peroxidation–antioxidant defense system” in various types of chronic gastritis in the presence of a serum response against *H. pylori* in the form of “IgG > IgA”.

The study found an increase in the median DC and MDA in patients with CG, CAG and CALG relative to the control group (Table 4). In addition, these indicators increased in patients with CALG compared with the group of CG and CAG.

Furthermore, the state of the AOD system in the groups of patients was assessed. It was found that the median values of superoxide dismutase, catalase activity, glutathione-S-transferase and glutathione peroxidase in plasma increased in the groups of patients with CAG and CALG compared with the control group. In addition, diene conjugates, malondialdehyde, and superoxide dismutase increased in patients with CALG compared with the CG and CAG groups, and the activity of catalase, glutathione-S-transferase, and glutathione peroxidase increased in patients with CALG compared with CG.

The coefficient of oxidative stress reflects the balance of the processes of lipid peroxidation from the antioxidant system and normally tends to be 1. When calculating the coefficient of oxidative stress with a serum response of IgG > IgA antibodies, it was revealed that: with chronic gastritis, OSR is 1.2; with CAG it is 3.14, and with CALG it is 2.1. In general, we can state a slight predominance of lipid peroxidation processes over antioxidant function, while more significant results are revealed in chronic atrophic gastritis, which once again explains the pathogenetic chain of the Korrea cascade in *H. pylori* infection.

#### 3.3.2. The State of the System “Lipid Peroxidation–Antioxidant Defense System” in Various Types of Chronic Gastritis in the Presence of a Serum Response against *H. pylori* in the Form of “Low Concentrations of IgG and IgA”

We assessed the state of the system “lipid peroxidation–antioxidant defense system” in various types of chronic gastritis in the presence of a serum response against *H. pylori* in the form of “low concentrations of IgG and IgA”.

The study found an increase in the median DC and MDA in patients with CG, CAG and CALG relative to the control group (Table 5). In addition, these indicators increased in patients with CALG compared with the group of CG and CAG.

Further, the state of the AOD system in the groups of patients was assessed. It was found that the median values of superoxide dismutase, glutathione-S-transferase, and glutathione peroxidase decreased in the group of patients with CAG and CALG compared with the control group. In patients with CALG, there is a decrease in these indicators compared with patients with chronic gastritis. The median of the catalase enzyme decreased in all groups of patients compared with the control group, in patients with CAG and CALG, there is a decrease in this indicator compared with the CG group.

When calculating the coefficient of oxidative stress for low concentrations of antibodies IgG, IgA, it was revealed that: with chronic gastritis, OSR is 1.2; with CAG it is 7.6, and with CALG it is 2.7. Low concentrations of antibodies indicate a reduced activity of the immune system cells, a low antibacterial immune response and a possible immunodeficiency state.

#### 3.3.3. The State of the System “Lipid Peroxidation–Antioxidant Defense System” in Various Types of Chronic Gastritis in the Presence of a Serum Response against *H. pylori* in the Form of an Increase in IgE

We assessed the state of the “lipid peroxidation–antioxidant defense system” in various types of chronic gastritis in the presence of a serum response against *H. pylori* in the form of an increase in IgE.

The study found an increase in the median DC and MDA in patients with CG, CAG and CALG relative to the control group (Table 6). In addition, these indicators increased in patients with CAG and CALG compared with the CG group, as well as in patients with CALG compared with CAG.

Further, the state of the AOD system in the groups of patients was assessed. It was found that the median values of superoxide dismutase, catalase increased in patients with chronic gastritis compared with the control group. In patients with CAG and CALG, there was an increase in superoxide dismutase, catalase, glutathione-S-transferase and glutathione peroxidase in plasma compared with the control group and patients with chronic gastritis. In patients with CALG, there was an increase in superoxide dismutase and catalase compared with CAG.

When calculating the coefficient of oxidative stress for high concentrations of IgE antibodies, it was revealed that: with chronic gastritis, OSR is 0.8; with CAG it is 0.76 and with CALG it is 0.56. In general, we can state a slight predominance of lipid peroxidation processes over the antioxidant function, with the most significant results being detected in chronic superficial gastritis. Probably, this change is due to the duration of the inflammatory process in the gastric mucosa. An increase in the concentration of IgE is associated not only with the presence of *H. pylori*, but also with the appearance of an additional allergen.

## 4. Discussion

An increasing number of studies confirm the close relationship between *H. pylori* infection and CG, as described previously. However, the correlation between *H. pylori* infection, serum antibodies and indicators of the “lipid peroxidation–antioxidant system” in various types of CG is still not well understood. The results of the chi-square test in Table 1 show that the identified positive rates of *H. pylori* are significantly increased in patients with superficial CG (78%, 47/60), CAG (91%, 30/33), CALG (83%, 25/6). They reach the greatest value in patients with CAG (91%, 30/33) (*p* < 0.01).

It has been reported that the attachment of *H. pylori* may play an important role in the pathogenesis of severe histological changes in CG and CAG [8,9]. In our study, we additionally studied the statistical probability of *H. pylori* infection in different types of chronic gastritis. Based on this, we suggest that *H. pylori* infection may be a key factor in accelerating the onset and development of CG, since *H. pylori* is a pathogenic bacterium that can attach to the gastric mucosa up to atrophy and intestinal metaplasia that occurs during its persistence and chronic inflammatory disease processes.

The most pathological mechanism of *H. pylori* infection is the development of an immunopathological host response. When CG occurs, a specific humoral immunological response is determined. The appearance of serum antibodies such as IgG and IgA may indicate an extensive immunoreaction caused by *H. pylori* infection. Thus, serum anti-*H. pylori*-IgG antibodies act as a highly accurate, simple and non-invasive method for monitoring the status of *H. pylori* infection [5,10]. In our study, we studied three variants of the immune response to *H. pylori* infection: a serum response in which specific IgG > IgA, with a low concentration of IgG and IgA, and a variant with an increase in IgE content. The results of the Chi-square test in Table 2 show that the humoral immune response in the form of specific IgG > IgA was detected in patients with superficial CG (75%, 45/60), CAG (60%, 19/33), CALG (37%, 12/31), (*p* < 0.01), with low concentrations of IgG and IgA; in patients with superficial CG (15%, 9/60), CAG (27%, 9/33), CALG (39%, 13/31) (*p* < 0.01) and a variant with an increase in the content of IgE; and in patients with superficial CG (10%, 6/60), CAG (13%, 4/33), CALG (100%, 31/31) (*p* < 0.01). We found in CG and CAG, the dominant variant of the immune response to *H. pylori* infection in the form of a serum response, in which specific IgG > IgA, and in CALG, with an increase in specific IgE (*p* < 0.01). An increase in the number of antibodies may be associated with the reaction of the humoral immunity to *H. pylori*, which leads to a certain imbalance between serum antibodies and *H. pylori*. It has been reported that the IgA response of the gastric mucosa in patients with CG can be detected even during the period of negative *H. pylori* infection due to recent exposure to bacterial antigens [6].

We evaluated the activity of the “lipid peroxidation–antioxidant system” in various forms of CG, depending on the variants of the immune response to *H. pylori* infection. With serum responses “antibodies IgG > IgA” and with “high concentrations of IgE antibodies”, we found unidirectional changes in the form of an increase in the amount of diene conjugates, malondialdehyde and an increase in the activity of all enzymes: superoxide dismutase, catalase, glutathione-S-transferase and glutathione peroxidase. In the first response variant, the most pronounced shift toward pro-oxidant processes was detected in CAG. Currently, there are a number of works confirming the importance of oxidative stress in the development of gastritis [11,12,13]. *H. pylori* strains containing certain subtypes of CagA and VacA are characterized by increased survival and ability to adhere, which increases damage to gastric epithelial cells and makes it possible to evade the immune response of the macroorganism [14]. It is known that CagA strains induce higher levels of hydrogen peroxide (H_2_O_2_) production and oxidative DNA damage [15]. The production of reactive oxygen species (ROS) and DNA damage is carried out in the epithelial cells of the stomach. *H. pylori* CagA induces the expression of spermine oxidase (SMO), which leads to the formation of H_2_O_2_, accumulation of ROS, and epithelial cell apoptosis. SMO is an enzyme of the polyamine metabolic pathway responsible for the reverse conversion of spermine to spermidine. At the same time, SMO is directly responsible for the generation of H_2_O_2_, which causes depolarization of the mitochondrial membrane and activation of caspase-mediated apoptosis [15,16].

Under oxidative stress, an excess of ROS in cells can damage tissues, leading to oncogenesis, especially in the gastrointestinal tract. This type of humoral response, in which specific IgG > IgA is a classic option, and in which it has already been proven by numerous studies to be ineffective. In the second variant (with an increase in IgE), despite the high concentrations of antibodies–reagins, the processes of lipid peroxidation were balanced by the activity of the antioxidant system, which allowed us to assume that the appearance of an additional immunogen (an allergen is a “stress” to the cells of the immune system) causes their systemic activation with the development of effective immune responses resulting in clinical improvement.

With a serum response with low concentrations of IgG, IgA antibodies, multidirectional changes were found in the form of an increase in the amount of diene conjugates, malondialdehyde and a decrease in the activity of all enzymes: superoxide dismutase, catalase, glutathione-S-transferase and glutathione peroxidase relative to the control group. In this case, the defective work of the immune apparatus was traced; there is even the possibility of an immunodeficiency state. A reduced immune response leads to the activation of *H. pylori*, an infection that causes oxidative stress and suppresses the body′s antioxidant function.

## 5. Conclusions

Thus, the onset and further progression of CG are associated with *H. pylori* infection, which may be characterized by an increase in serum anti-*H. pylori* IgG and IgA, IgE or with some decrease in them and with the development of oxidative stress. The balance of lipid peroxidation processes and the antioxidant system depend on the characteristics of the immune response to *H. pylori* infection. Interesting data have been obtained on LPO–AOD parameters in the case of high IgE content in various types of CG, which require further study. The presence of antibodies to *H. pylori* contributes to the development of immune responses by the Th 2 mechanism. Immune complexes, phagocytosis, and non-specific and specific immune reactions exacerbate inflammation in the gastric mucosa, destroying epitheliocytes and thereby triggering the process of lipid peroxidation and oxidative stress. There is a direct relationship between the severity of the immune response and the degree of oxidative stress. Therefore, in order to reduce the manifestations of lipid peroxidation of cell membranes, it is necessary to eliminate immune inflammation, which will decrease after the eradication of *H. pylori* in the gastric mucosa.

## Figures and Tables

**Figure 1 biomedicines-10-02053-f001:**
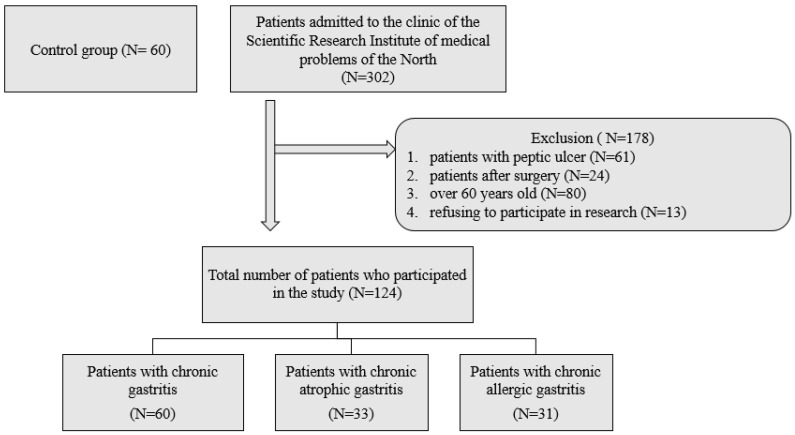
Scheme of inclusion of patients in the study.

**Figure 2 biomedicines-10-02053-f002:**
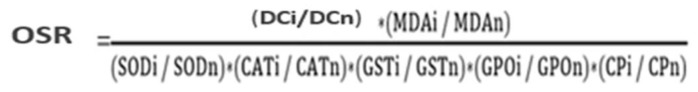
Formula for calculating OSR.

**Table 1 biomedicines-10-02053-t001:** Baseline characteristics of 124 subjects with various types of gastritis infected with *Helicobacter pylori*.

Variable	CG, N = 60	CAG, N = 33	CALG, N = 31	*p* Value
Age, yr	47.54 ± 12.65	57.3 ± 9.65	49.87 ± 8.34	0.414
Body mass index, kg/m^2^	23.78 ± 2.62	22.41 ± 2.45	22.31 ± 3.68	0.237
Alcohol consumption 1. Never 2. Past 3. Current				0.468
	10 (15.3)	8 (10.9)	4 (13.8)	
	7 (17.4)	6 (20.7)	3 (11.8)	
	43 (71.7)	19 (65.5)	24 (82.4)	
Smoking 1. Never 2. Past 3. Current				0.842
	6 (34.8)	6 (29.4)	12 (41.4)	
	20 (43.5)	7 (47.1)	11 (37.9)	
	34 (21.7)	20 (23.5)	8 (20.7)	
pH value of gastric juice	3.23 ± 1.99	4.54 ± 2.23	2.46 ± 1.73	0.003
Serum pepsinogen content	N = 56	N = 29	N = 30	
Pepsinogen I, ng/mL	56.48 ± 89.13	93.16 ± 147.33	66.33 ± 93.25	0.133
Pepsinogen II, ng/mL	19.32 ± 28.44	34.80 ± 41.62	27.56 ± 38.67	0.068
Pepsinogen I/II ratio	3.68 ± 1.79	3.06 ± 1.61	3.49 ± 1.36	0.232

**Table 2 biomedicines-10-02053-t002:** *Helicobacter pylori* infection in various types of chronic gastritis (%/abs.).

*H. pylori*	CG	CAG	CALG	Total
Total number of patients	60	33	31	124
The presence of antibodies (%/abs.)	78%/47	91%/30	83%/25	102
Absence of antibodies, (%/abs.)	22%/13	9%/3	17%/6	22

**Table 3 biomedicines-10-02053-t003:** Characteristics of serum responses (antibodies IgG, IgA, IgE) against *H. pylori* in various types of chronic gastritis (%/abs.).

Serum Response Against *H. pylori*	CG	CAG	CALG	Total
IgG > IgA, (%/abs.)	75%/35	60%/18	5%/1	54
Low concentrations of IgG, IgA, (%/abs.)	15%/7	27%/8	4%/1	16
IgE, (%/abs.)	10%/5	13%/4	91%/23	32

**Table 4 biomedicines-10-02053-t004:** Characteristics of LPO–AOD system indicators in various types of chronic gastritis with serum response against *H. pylori* as “IgG > IgA”.

Indicators	Control Group, N = 43 (1)		CG, N = 47 (2)		CAG, N = 30 (3)		CALG, N = 25 (4)	
	Me	Q_1_–Q_3_	Me	Q_1_–Q_3_	Me	Q_1_–Q_3_	Me	Q_1_–Q_3_
DC, µmol/L	1.15	0.88–1.38	1.4	1.97–2.78	2.8	1.43–1.98	3.16	3.12–3.53
			*p_1–2_ =* 0.02		*p_1–3_ <* 0.001		*p_1–4_ <* 0.001; *p_2–4_ <* 0.001; *p_3–4_ <* 0.001	
MDA, µmol/1 g protein	1.6	0.96–2.24	2.1	1.42–2.8	2.5	1.6–2.9	3.3	2.6–4.7
			*p_1–2_* = 0.01		*p_1–3_* < 0.001		*p_1–4_ <* 0.001; *p_2–4_ <* 0.001; *p_3–4_ <* 0.001	
SOD, U/min/1 g protein	204.41	151.05–250.3	187.6	141.6–213.3	191.3	157.7–210.2	218.3	200.3–243.4
					*p_1–3_ =* 0.01		*p_1–4_* < 0.001; *p_2–4_ <* 0.001; *p_3–4_ <* 0.001	
CAT, µmol/s/1 g protein	0.2	0.16–0.39	0.18	0.12–0.29	0.25	0.19–0.43	0.36	0.27–0.64
					*p_1–3_* = 0.03		*p_1–4_ <* 0.001; *p_2–4_ <* 0.001	
GST, mmol/min/1 g protein	41.3	37.7–42.64	50.2	24.9–51.7	54.2	55.1–70.3	68.6	56.3–74.6
					*p_1–3_ =* 0.04		*p_1–4_ <* 0.001; *p_2–4_ <* 0.001	
GPO, µmol/1 g protein	107.9	81.19–126.38	105.02	68.9–122.1	118.6	132.1–176.5	147.4	138.1–194.4
					*p_1–3_ =* 0.04		*p_1–4_ = 0*.01; *p_2–4_* = 0.01	

Note: *p_1–2_*–significant differences between the control group and the group of patients with CG; *p_1–3_*–significant differences between the control group and the group of patients with CAG; *p_2–3_*–significant differences between the groups of CG and CAG; *p_1–4_*–significant differences between the control and CALG groups; *p_2–4_*–significant differences between the groups of CG and CALG; *p_3–4_*–significant differences between the groups of CAG and CALG.

**Table 5 biomedicines-10-02053-t005:** Characteristics of indicators of the LPO–AOD system in various types of chronic gastritis with a serum response against *H. pylori* in the form of “low concentrations of IgG, IgA”.

Indicators	Control Group, N = 43 (1)		CG, N = 47 (2)		CAG, N = 30 (3)		CALG, N = 25 (4)	
	Me	Q_1_–Q_3_	Me	Q_1_–Q_3_	Me	Q_1_–Q_3_	Me	Q_1_–Q_3_
DC, µmol/L	1.15	0.88–1.38	1.38	1.43–1.98	2.4	1.97–2.78	3.16	3.12–3.53
					*p_1–3_ <* 0.001; *p_2–3_ =* 0.001		*p_1–4_ <* 0.001; *p_2–4_ <* 0.001; *p_3–4_ <* 0.001	
MDA, µmol/1 g protein	1.6	0.96–2.24	2.4	1.9–3.1	3.1	2.9–4.8	4.8	3.6–6.7
					*p_1–3_ <* 0.001; *p_2–3_ =* 0.03		*p_1–4_ <* 0.001; *p_2–4_ <* 0.001; *p_3–4_ <* 0.001	
SOD, U/min/1 g protein	204.41	151.05–250.3	198.3	186.1–213.1	192.1	175–20.2	185.3	174.1–205.1
					*p_1–3_ =* 0.03		*p_1–4_ <* 0.001; *p_2–4_ =* 0.01	
CAT, µmol/s/1 g protein	0.2	0.16–0.39	0.19	0.13–0.2	0.15	0.12–0.22	0.14	0.12–0.24
			*p_1–2_ = 0.04*		*p_1–3_ <* 0.001; *p_2–3_ =* 0.03		*p_1–4_* < 0.001; *p_2–4_ =* 0.01	
GST, mmol/min/1 g protein	41.3	37.7–42.64	39.1	26.8–47.7	34.2	29.1–40.3	33.6	26.3–44.6
					*p_1–3_ =* 0.01		*p_1–4_ <* 0.001; *p_2–4_ =* 0.001	
GPO, µmol/1 g protein	107.9	81.19–126.38	109.3	91.3–117.3	98.6	89.1–106.5	97.4	91.1–104.4
					*p_1–3_ =* 0.001		*p_1–4_* < 0.001; *p_2–4_ =* 0.01	

see note in Table 4.

**Table 6 biomedicines-10-02053-t006:** Characterization of indicators of the LPO–AOD system in various types of chronic gastritis with a serum response against *H. pylori* in the form of an increase in “IgE”.

Indicators	Control Group, N = 43 (1)		CG, N = 47 (2)		CAG, N = 30 (3)		CALG, N = 25 (4)	
	Me	Q_1_–Q_3_	Me	Q_1_–Q_3_	Me	Q_1_–Q_3_	Me	Q_1_–Q_3_
DC, µmol/L	1.15	0.88–1.38	1.7	1.43–1.98	2.9	2.71–3.08	3.98	3.52–4.03
			*p_1–2_ =* 0.02		*p_1–3_ <* 0.001; *p_2–3_ =* 0.003		*p_1–4_ <* 0.001; *p_2–4_ <* 0.001; *p_3–4_ <* 0.001	
MDA, µmol/1 g protein	1.6	0.96–2.24	2.6	1.9–3.3	3.8	3.3–4.8	5.1	4.6–7.7
			*p_1–2_ =* 0.02		*p_1–3_ <* 0.001; *p_2–3_ =* 0.001		*p_1–4_ <* 0.001; *p_2–4_ <* 0.001; *p_3–4_ <* 0.001	
SOD, U/min/1 g protein	204.41	151.05–250.3	215.3	207.2–252.4	224.3	219.1–257.6	281.2	240.4–298.5
			*p_1–2_ =* 0.04		*p_1–3_ <* 0.001; *p_2–3_ =* 0.03		*p_1–4_ <* 0.001; *p_2–4_ <* 0.001; *p_3–4_ <* 0.001	
CAT, µmol/s/1 g protein	0.2	0.16–0.39	0.6	0.41–0.72	1.01	0.82–1.12	1.23	0.97–1.44
			*p_1–2_ =* 0.01		*p_1–3_ <* 0.001; *p_2–3_ =* 0.04		*p_1–4_ <* 0.001; *p_2–4_ <* 0.001; *p_3–4_ <* 0.001	
GST, mmol/min/1 g protein	41.3	37.7–42.64	53.5	46.8–67.7	64.2	55.1–87.3	78.6	61.3–94.6
					*p_1–3_ <* 0.001; *p_2–3_ =* 0.004		*p_1–4_ <* 0.001; *p_2–4_ <* 0.001	
GPO µmol/1 g protein	107.9	81.19–126.38	104.3	91.3–117.3	127.6	114.1–146.5	137.4	128.1–154.4
					*p_1–3_ =* 0.01; *p_2–3_ =* 0.004		*p_1–4_ <* 0.001; *p_2–4_ <* 0.001	

see note in Table 4.

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
