# Peer review of "The Role of Immunoglobulin G (IgG), IgA and IgE—Antibodies against Helicobacter pylori in the Development of Oxidative Stress in Patients with Chronic Gastritis"

_biomedicines, 2022, doi:10.3390/biomedicines10082053_

Round 1
Reviewer 1 Report
The manuscript entitled “The Role of Immunoglobulin G (IgG), IgA and IgE -Antibodies against Helicobacter Pylori in the Development of Oxidative Stress in Patients with Chronic Gastritis” describes that with serum responses “antibodies IgG>IgA” and “high concentrations of IgE antibodies”, authors found unidirectional changes in the form of an increase in the amount of diene conjugates, malondialdehyde and an increase in the activity of all enzymes: superoxide dismutase, catalase, glutathione-S-transferase and glutathione peroxidase. With a serum response with low concentrations of IgG, IgA antibodies, multidirectional changes were found in the form of an increase in the amount of diene conjugates, malondialdehyde and a decrease in the activity of all enzymes: superoxide dismutase, catalase, glutathione-S-transferase and glutathione peroxidase relative to the control group.
This study provides the association between immunoglobulin antibodies and the peroxidation/antioxidant system processes. Authors should further prove the causal relationship.
Author Response
The presence of antibodies to H. pylori contributes to the development of immune responses by the Th 2 mechanism. Immune complexes, phagocytosis, non-specific and specific immune reactions exacerbate inflammation in the gastric mucosa, destroying epitheliocytes and thereby triggering the process of lipid peroxidation and oxidative stress. There is a direct relationship between the severity of the immune response and the degree of oxidative stress. Therefore, in order to reduce the manifestations of lipid peroxidation of cell membranes, it is necessary to eliminate immune inflammation, which will decrease after the eradication of H. pylori in the gastric mucosa.
Reviewer 2 Report
The Role of Immunoglobulin G (IgG), IgA and IgE - Antibodies against Helicobacter Pylori in the Development of Oxidative Stress in Patients with Chronic Gastritis. by Olga Valentinovna Smirnova et al.
To the Authors:
General comments:
The authors investigated the predominant serum responses (antibodies IgG, IgA, IgE) against H. pylori about the indicators of the system "lipid peroxidation - antioxidant system" in various pathogenetic variants of chronic gastritis. They found unidirectional changes in the form of an increase in the amount of diene conjugates, malondialdehyde, and an increase in the activity of enzymes with serum responses “antibodies IgG>IgA” and “high concentrations of IgE antibodies.” It was considered that the study was well structured, and the result was clearly written. Also, the result included novelty. However, several points should be addressed to improve the manuscript.
Specific comments:
1. Did the patients use drugs including proton pump inhibitors and non-steroidal anti-inflammatory drugs, which may affect the degree of gastritis? Please show the data regarding their history of drug use.
2. Patients with gastritis caused by H. pylori may be asymptomatic. What were the results of the endoscopic examination for the control group?
3. The authors stated that “All patients included in the study were infected with Helicobacter pylori (Line 57-58).” On the other hand, in line 187-190, they described that “In patients with chronic gastritis, the presence of H. pylori infection was detected in 78% of cases, in patients with CAG, the presence of this infection was detected in 91% of cases, and in patients with CALG, the presence of H. pylori was 83%.” This discrepancy seems confusing. Please clarify this point.
Minor comments:
1. The authors should spell out CG mentioned the first time, not the second time, in the Abstract.
Author Response
- The study included only primary patients with a verified diagnosis without a history of the use of proton pump inhibitors and non-steroidal anti-inflammatory drugs for the purity of the experiment.
- The results of endoscopic examination of persons in the control group revealed infection with H. pylori in 71% of those examined with asymptomatic course.
- We obtained a different prevalence of H. pylori infection in different types of gastritis in primary patients and in the control group. Subsequently, only individuals with H. pylori infection, proven by endoscopic and serological methods, were taken into the study.
- Bugs fixed.
Round 2
Reviewer 1 Report
Authors have well revised this manuscript based on suggestions and comments of reviewers.
Reviewer 2 Report
To the Authors:
General comments: The authors revised the manuscript according to the comments appropriately.